

# Using virtual reality for anatomical landmark annotation in geometric morphometrics

Dolores Messer[1,*], Michael Atchapero[2,*], Mark B. Jensen[1],
Michelle S. Svendsen[3], Anders Galatius[4], Morten T. Olsen[3],
Jeppe R. Frisvad[1], Vedrana A. Dahl[1], Knut Conradsen[1], Anders B. Dahl[1]
and Andreas Bærentzen[1]

[1] Department of Applied Mathematics and Computer Science, Technical University of Denmark, Kgs. Lyngby, Denmark
[2] Department of Psychology, University of Copenhagen, Copenhagen, Denmark
[3] Globe Institute, University of Copenhagen, Copenhagen, Denmark
[4] Department of Ecoscience, Aarhus University, Roskilde, Denmark
* These authors contributed equally to this work.

Corresponding author
Dolores Messer, dolmes@dtu.dk

## ABSTRACT

To study the shape of objects using geometric morphometrics, landmarks are oftentimes collected digitally from a 3D scanned model. The expert may annotate landmarks using software that visualizes the 3D model on a flat screen, and interaction is achieved with a mouse and a keyboard. However, landmark annotation of a 3D model on a 2D display is a tedious process and potentially introduces error due to the perception and interaction limitations of the flat interface. In addition, digital landmark placement can be more time-consuming than direct annotation on the physical object using a tactile digitizer arm. Since virtual reality (VR) is designed to more closely resemble the real world, we present a VR prototype for annotating landmarks on 3D models. We study the impact of VR on annotation performance by comparing our VR prototype to Stratovan Checkpoint, a commonly used commercial desktop software. We use an experimental setup, where four operators placed six landmarks on six grey seal (*Halichoerus grypus*) skulls in six trials for both systems. This enables us to investigate multiple sources of measurement error. We analyse both for the configuration and for single landmarks. Our analysis shows that annotation in VR is a promising alternative to desktop annotation. We find that annotation precision is comparable between the two systems, with VR being significantly more precise for one of the landmarks. We do not find evidence that annotation in VR is faster than on the desktop, but it is accurate.

## INTRODUCTION

Geometric morphometrics is a powerful approach to study shape and is widely used to capture and quantify shape variation of biological objects such as skulls (*Rohlf & Marcus, 1993*; *Bookstein, 1998*; *Adams, Rohlf & Slice, 2004*; *Slice, 2005*; *Mitteroecker & Gunz,*

*2009*). In geometric morphometrics, shapes can be described by locating points of correspondences in anatomically meaningful landmark positions that are easily identifiable (*Bookstein, 1991*). An acknowledged and widely used practice is to obtain landmarks directly from a physical specimen through a tactile 3D digitizer arm (*Sholts et al., 2011*; *Waltenberger, Rebay-Salisbury & Mitteroecker, 2021*). However, collecting landmarks digitally from a 3D scanned model of the physical specimen is also a viable and widely accepted alternative that has found increased use in recent times (*Sholts et al., 2011*; *Robinson & Terhune, 2017*; *Bastir et al., 2019*; *Messer et al., 2021*; *Waltenberger, Rebay-Salisbury & Mitteroecker, 2021*). Annotation of a digital 3D model is typically done using a software tool based on mouse, keyboard, and 2D display (*Bastir et al., 2019*), and hence placing landmarks can be a tedious task since the perception of shape in a 2D environment is limited as compared with the real world. When a landmark is placed, *e.g.*, a landmark on a 3D tip, small rotations are needed to verify the position, otherwise there might be a significant distance between the actual and the desired landmark coordinates. Annotation of 3D landmarks on a 2D display is more time-consuming than when using a digitizer arm (*Messer et al., 2021*). On the other hand, having landmarks placed on a 3D scan of a model carries a number of advantages in terms of data sharing and repositioning of landmarks compared to stand-alone landmarks from a 3D digitizer (*Waltenberger, Rebay-Salisbury & Mitteroecker, 2021*). We argue that virtual reality (VR) provides a closer-to-real-world alternative to desktop annotation that retains the multiple benefits of having the landmarks on a 3D scanned model, including the ability to easily share the digital 3D model, examine it from all angles and accurately place landmarks. The user interaction afforded by the VR head-mounted display allows navigators to move the virtual camera while the controllers can move and rotate the object and serve as an annotation tool as well. All in all, the head-mounted display and controllers exhibit six degrees of freedom (DOF), which map directly to the six DOF needed to intuitively navigate in a 3D environment. This should significantly ease the annotation process and hence make 3D models more useful in biological studies that often require large sample sizes to obtain robust statistics. To investigate the use of VR for digitally annotating landmarks on animal 3D models, we present a prototype VR annotation system and study the impact of VR on annotation performance as compared with a traditional system using 2D display and user interaction by mouse and keyboard.

When comparing a desktop interface to a VR interface, some aspects of VR should be considered. Both latency and tracking noise is higher in VR than with a standard computer mouse. This can degrade performance and precision (*Teather et al., 2009*). Furthermore, most VR controllers are held in a power grip (clutching the fingers around the object, thereby using the strength of the wrist), as opposed to holding a mouse in a precision grip (holding an object with the fingertips, such as when using a pen, thereby enabling the finer motor skills of the fingers). This makes it more difficult to be precise when annotating objects using most VR controllers (*Pham & Stuerzlinger, 2019*). Using the mouse on the other hand allows for more controlled and more precise movements, all while allowing the user to let go of the mouse without loosing the position on the 3D

model. While the mouse only has three DOF that need to be mapped to the six DOF, it still allows for direct manipulation of either the object or the camera.

The benefits of digital 3D representation of biological specimens (such as skulls) was discovered more than two decades ago (*Recheis et al., 1999*). This developed into the more inclusive field of digital or virtual morphology (*Weber, 2015*), and the workflows in a virtual morphology lab is now a topic of considerable interest (*Bastir et al., 2019*). *Bastir et al. (2019)* discuss the various databases and the key software tools available for geometric morphometrics. One of the discussed software tools is Landmark editor (*Wiley et al., 2005*), which is the predecessor of Stratovan Checkpoint. It seems that none of the software tools in this area employ virtual reality.

VR allows an operator to virtually annotate landmarks in 3D models in a way that resembles real-world annotation of physical specimens (*Bowman, McMahan & Ragan, 2012*; *Mendes et al., 2019*). The directness of this interaction produces a short distance between thought and physical action, making for a simple and straightforward interaction modality. More direct interaction demands a lower cognitive load (*Hutchins, Hollan & Norman, 1985*). More cognitive effort can then be invested in understanding and interacting with the data that are presented. *Jang et al. (2017)* showed that direct manipulation in VR provides a better understanding, and that it benefited students with low spatial ability the most. *Bouaoud et al. (2020)* found that students gain a better understanding of craniofacial fractures by inspecting 3D models based on CT scans in VR.

In a recent study, *Cai et al. (2020)* found that using VR to teach about deformities in craniovertebral junctions would improve the ability of the students when afterwards placing landmarks in radiographs of craniovertebral junctions with deformities. This was an improvement as compared with students receiving teaching with physical models. We consider this an indicator that perhaps the act of placing landmarks in digital morphology could be improved too if performed in VR. We follow up on this indication and compare precision and accuracy in placing landmarks on virtual 3D representations of skulls when using our VR system and when using the traditional 2D display and mouse interface.

The possibility of *haptic* feedback is an important aspect of VR input devices. Haptic pertains to the sense of touch, and we can broadly distinguish between two types of haptic feedback. Purely tactile feedback simply means that nerves in your skin are stimulated when you touch something. Standard VR controllers support this through the expedient of vibration, and this is often called *vibrotactile* feedback. The word *kinesthetic* is used about the sense of how limbs of a person's body are positioned in space. Thus, a device which provides force feedback, thereby preventing your hand from going through a virtual surface, is often described as kinesthetic.

Within the area of placing medical landmarks, *Li et al. (2021)* performed a comparison of the traditional 2D display and mouse interface with two variants of the VR interface, one using standard VR controllers held in a power grip, and one using kinesthetic controllers held in a precision grip. Note that this study differs from ours in another important respect: They show markers which the participants are supposed to target when annotating, whereas we consider the task of deciding where to place the point to be integral

to the annotation task (*i.e.* there is no ground truth). The improvement in marking accuracy was found to be statistically significant when the kinesthetic input device was employed. The task completion time and difficulty of use was however higher for the kinesthetic VR device as compared with the standard vibrotactile controller. The latter was thus easier to use and had as good overall accuracy as the 2D display and mouse interface. Interestingly, the results of *Li et al. (2021)* suggest that marking accuracy in VR is less affected by marking difficulty than when using a 2D interface. The task performance was thus more stable in VR than when using the traditional 2D tool. This is another motivation behind our test of the performance of placing landmarks in VR as compared with a traditional 2D tool. *Li et al. (2021)* show that when mouse and VR interfaces are used in a similar way, the haptic feedback helps improving marking accuracy. They do so by having the users interact with the virtual world through an asymmetric bimanual (*i.e.* using both hands) interface, where one hand holds the controller or mouse which is used to both manipulate and annotate the virtual objects, while the other hand can press the spacebar to place the annotation point. Because the same hand is used both for manipulation and marker placement, their interface does not follow the theoretical framework for designing an asymmetric bimanual interface by *Guiard (1987)*. *Kabbash, Buxton & Sellen, 1994* show that carefully designed asymmetric bimanual interfaces can improve task performance, while inappropriately designed interfaces lower performance. *Balakrishnan & Kurtenbach (1999)* show that using an asymmetric bimanual interface designed with the theoretical framework proposed by *Guiard (1987)* leads to a 20% performance increase over a unimanual interface. Our study employs an asymmetric bimanual interface that follows the theoretical framework by *Guiard (1987)*, and investigates whether combining this interface with regular VR controllers will lead to similar improvements in annotation performance, saving the users from having to acquire special purpose haptic devices.

In geometric morphometrics studies, the presence of measurement error can influence the results of the performed analysis by increasing the level of noise, which can obscure the biological signal, and/or by introducing bias (*Fruciano, 2016*). *Fruciano (2016)* discusses the different sources of measurement error. Several studies in geometric morphometrics quantified measurement error in a situation where landmark data were collected using different devices, and 3D capture modalities (*e.g.* micro CT, surface scanner, photogrammetry), involving several operators (*Robinson & Terhune, 2017*; *Shearer et al., 2017*; *Fruciano et al., 2017*; *Giacomini et al., 2019*; *Messer et al., 2021*). Different systems for annotation of digital 3D models were however not compared in these studies.

To investigate whether annotation in VR is a viable alternative to mouse and keyboard for digital annotation of landmarks on 3D models, we compare our VR prototype to Stratovan Checkpoint (Stratovan Corporation, Davis, CA, USA; https://www.stratovan.com/products/checkpoint), a commonly used software for digitally annotating landmarks on 3D models using mouse and keyboard. We note that Stratovan Checkpoint comes with many features, but we focus only on the placing of anatomical landmarks. We study the impact of VR on annotation performance. In a first step, we assess overall and landmark-wise precision and accuracy. Moreover, we investigate different sources of

measurement error (between systems, between and within operators) in an overall and landmark-wise explorative analysis. Finally, we investigate differences in annotation time between systems and operators, and over time.

# MATERIALS AND METHODS

## VR annotation system

The VR annotation system was developed at the Technical University of Denmark using Unity 2019 (Unity Technologies, San Francisco, CA, USA; https://unity.com) and the Oculus Rift hardware released in 2016 (Facebook Technologies, LLC, Menlo Park, CA, USA; https://www.oculus.com). Users in the virtual environment are presented with an asymmetric bimanual interface, which adheres to *Guiard (1987)*'s three high-order principles: Assuming a right-handed subject, (1) The system uses a *right-to-left reference* where motion of the right hand finds its spatial reference relative to the left hand. The user manipulates and orients the skull by grabbing it with the left controller. The skull can be scaled by pressing buttons on the left controller. The right controller is then used to place the annotation point, this is done with a ray-gun. The ray-gun shoots out a virtual red laser-line that is intersecting with the surface of the 3D model. Landmark annotation is mimicking the gesture of shooting a gun: The user aims by pointing the controller at the landmark location, and presses the index trigger of the controller. An annotation gun is chosen since it fits well with the power grip that the VR controllers are held in. (2) The actions of the left and right hand are on *asymmetric scales of motion* where the left hand performs large scale movements adjusting the skull and the right hand performs small scale movements to set the annotation point. (3) This workflow means that the left hand moves before the right hand, adhering to the principle of *left-hand precedence*. The VR annotation system supports both right- and left-handed subjects.

3D models are rendered opaque. It is possible to move the viewpoint such that the inside of a 3D model is visible. As opposed to the desktop software Stratovan Checkpoint, the inside of a 3D model is not rendered. Only 3D models are rendered, with no additional information being shown, *i.e.* unlike other systems, we do not show cross sections.

A comprehensive description of the VR annotation system, and a detailed comparison of the VR system to the desktop software Stratovan Checkpoint and a 3D digitizer arm are provided in the Article S1. A demonstration of the VR annotation system, and Stratovan Checkpoint, are shown in Videos S1 and S2, respectively.

## Landmark data collection

Our study investigates how precisely, accurately and fast a user can place landmarks using our VR annotation system compared to Stratovan Checkpoint, a mouse and keyboard based desktop system. To carry out this experiment, we scanned, reconstructed, and annotated grey seal skulls.

### Sample

The sample consisted of six grey seal (*Halichoerus grypus*) skulls. Of these, five skulls were held by the Natural History Museum of Denmark (NHMD) and originated from the Baltic

Sea population, whereas one skull originated from the western North Atlantic population and was held by the Finnish Museum of Natural History in Helsinki (FMNH) (Table A1). We selected the skulls based on size in order to cover a large span: In our sample, skull length ranges from about 18 to 28 cm. All the selected specimens were intact, and did not have any abnormalities in size, shape or colour variation. We did not include mandibles.

### Generation of 3D models

The 3D models of the specimens were generated as previously described in *Messer et al. (2021)*. In a first step, the skulls were 3D scanned in four positions using a 3D structured light scanning setup (SeeMaLab (*Eiriksson et al., 2016*)). On the basis of geometric features, the point clouds from the four positions of a given skull were then globally aligned using the Open3D library (*Zhou, Park & Koltun, 2018*), followed by non-rigid alignment as suggested by *Gawrilowicz & Bærentzen (2019)*. The final 3D model was reconstructed on the basis of Poisson surface reconstruction (*Kazhdan, Bolitho & Hoppe, 2006*; *Kazhdan & Hoppe, 2013*) using the Adaptive Multigrid Solvers software, version 12.00, by Kazhdan (Johns Hopkins University, Baltimore, MA, USA; https://www.cs.jhu.edu/~misha/Code/PoissonRecon/Version12.00).

In a last step, we used the Decimate function with a ratio of 0.1 in the Blender software, version 2.91.2, (Blender Institute B.V., Amsterdam, the Netherlands; https://www.blender.org) to downsample the final meshes of all specimens, thereby reducing the number of faces to about 1.5 million. The average edge length, which is a measure of 3D model resolution, is between 0.312 mm (smallest skull) and 0.499 mm (largest skull). Original meshes consist of about 15 million faces, and we performed downsampling to ensure that our hardware could render the meshes with a frame rate suitable for VR. By using the same resolution 3D model for both the VR and traditional desktop system, we ensured that observed differences in precision and accuracy were not due to differences in resolution. Figure S1 shows the six final 3D models. A comparison of the original with the downsampled 3D model is illustrated in Fig. S2 using FMNH specimen C7-98.

### Annotation of landmarks

For each of the six skulls, Cartesian coordinates of six fixed anatomical landmarks (Fig. 1; Table A2) were recorded by four operators. Each operator applied two different systems to place the landmarks on the reconstructed digital 3D models of the grey seal skulls: (1) Stratovan Checkpoint software, version 2018.08.07, (Stratovan Corporation, Davis, CA, USA; https://www.stratovan.com/products/checkpoint), without actively using the simultaneous view of three perpendicular cross-sections, and (2) our own virtual reality tool (Article S1). To assess within-operator error, each operator annotated the same skull six times with both systems. In total, 288 landmark configurations were collected.

Our choice of landmarks is a subset of six out of 31 previously defined anatomical landmarks on grey seal skulls (*Messer et al., 2021*). We chose the landmarks with indices 2, 3, 11, 18, 24, 28 to have landmarks both of Type I (three structures meet, *e.g.* intersections of sutures) and Type II (maxima of curvature) (*Bookstein, 1991*; *Brombin & Salmaso,*

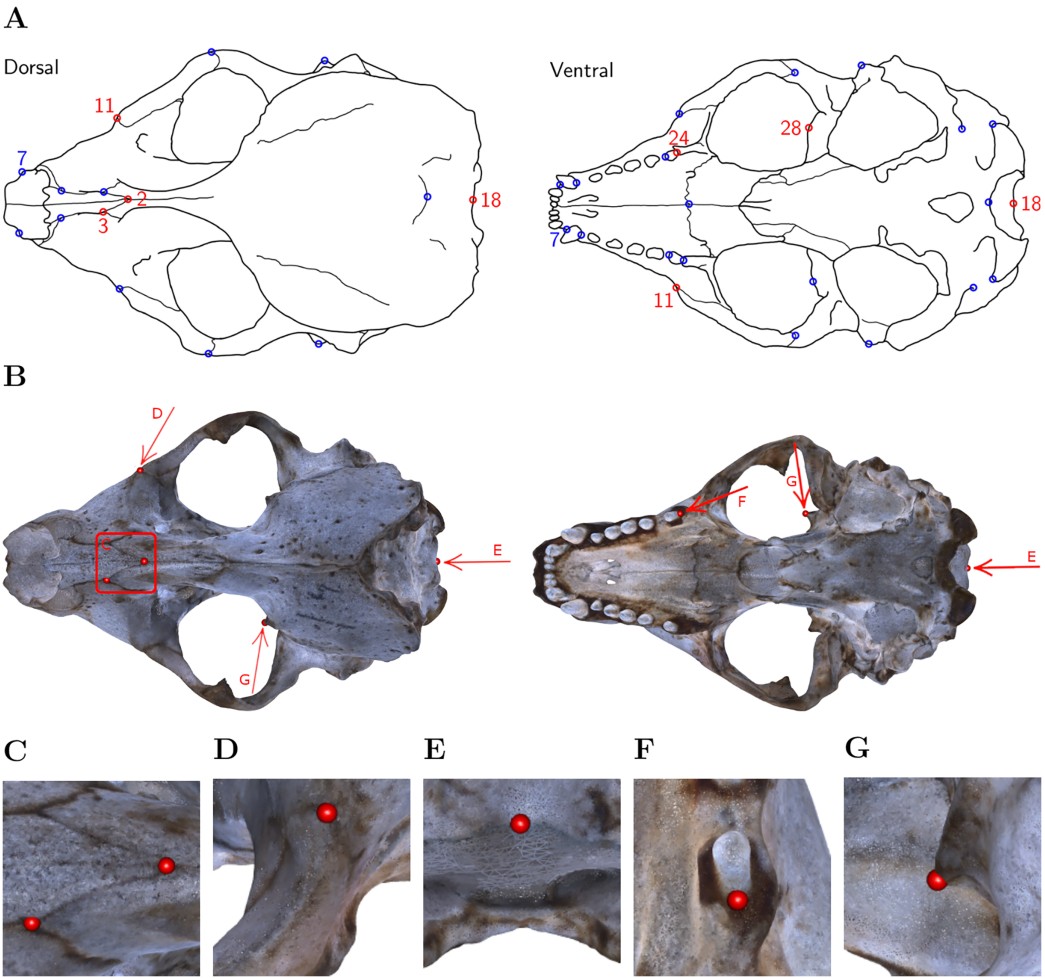

**Figure 1 Landmark definition.** (A) For our study, we selected the six landmarks marked in red based on the set of 31 anatomical landmarks on grey seal skulls as defined by *Messer et al. (2021)*. *Note:* Adapted from *Messer et al. (2021)*. Reprinted with permission. (B) The six landmarks placed on the 3D model of NHMD specimen 96. Square/Arrows indicate the camera view for taking images (C)–(G), which show the placed landmarks on the 3D model of NHMD specimen 96 and highlight their respective features. (C) Landmarks 2 and 3: Intersections of sutures; (D) Landmark 11: Apex along a margin; (E) Landmark 18: Medial point of a margin; (F) Landmark 24: Posterior/saddle point; (G) Landmark 28: 3D tip.

*2013*). Moreover, we excluded symmetric landmarks, and landmarks not well defined on all skulls. We selected landmarks located at points that are spread over the whole skull while exhibiting different characteristics.

Experience in placing landmarks, and experience in annotating digital models are two important factors that are likely to influence operator measurement error. Thus, our chosen operators differ from each other with respect to these two relevant factors: Two operators (A and D) were biologists, both of them having experience annotating landmarks using a Microscribe® digitizer, but only operator D had experience placing landmarks on 3D models using Stratovan Checkpoint. The other two operators (B and C) had a background in virtual reality. Operator B had previously annotated landmarks on

one grey seal specimen in both systems and was the developer of the virtual reality annotation tool presented in this study. Operator C was the only one having no experience in placing landmarks and was collecting the landmark data in a test run prior to other data collection. Operators A, B and D spread data collection over 2 to 3 days, whereas operator C annotated all 3D models on the same day.

All operators recorded the landmark configurations in the same pre-defined, randomized order (Table S1), making them switch between systems and specimens. The primary goal was to prevent the operators from memorizing where they previously had placed the landmarks on a particular skull using a specific system. Operator A slightly changed the order by swapping NHMD specimens 223 and 96 using the virtual reality tool, and NHMD specimens 323 and 664 using Stratovan Checkpoint, both for the third replica. Moreover, Operator A accidentally skipped specimen 323 once (virtual reality, third replica) during data collection, and thus annotated this skull 3 months later. In each annotation round, operators sequentially placed the landmarks in the order (2, 3, 11, 18, 24, 28). Operator C, however, collected landmark coordinates in a different order (18, 2, 3, 24, 28, 11).

In addition to collecting landmark coordinates, the annotation time was recorded for each measurement of six landmarks. In case of the virtual reality tool, annotation time was automatically recorded, whereas the operators had to manually record the time using a small stopwatch program that ran in a console window when they annotated landmarks in Stratovan Checkpoint. The operators were instructed to use as much time as they needed for a satisfactory annotation.

## Statistical data analysis and outliers

All statistical analyses were conducted in the R software version 3.5.3 (*R Core Team, 2020*). For geometric morphometric analyses, we used the package geomorph (*Adams & Otárola-Castillo, 2013*).

There were three different types of outliers present in the raw landmark coordinates data: (1) swapped landmarks, (2) obviously wrongly placed landmarks (4–9 mm away from all corresponding replicas; mean Euclidean distance between corresponding replicas was 0.02–0.31 mm)[1], and (3) landmarks localized at two distinct points, for several replicas at each point, or distributed between two distinct points. We put swapped landmarks into the correct order, and replaced the two obviously wrongly placed landmarks by an estimate based on the remaining five replicas using geomorph's function estimate.missing (thin-plate spline approach). There were two cases of outliers of type (3): Landmark 18 annotated by operator C on FMNH specimen C7-98, and landmark 28 annotated by operator A on NHMD specimen 42.11, in both cases when using Stratovan Checkpoint as well as the VR annotation system (Fig. S3). Assuming that in these two cases, the operators were in doubt where to clearly place the landmarks, we decided to include outliers of type (3) in all our analyses to not confound the results.

In our design, the repetitions were performed in a randomized order to avoid obvious sources of autocorrelations between repeated measurements on the same landmarks. This

[1] The two outliers of Type (2) were replica 3 of landmark 11 on NHMD specimen 664, and replica 6 of landmark 28 on FMNH specimen C7-98, both placed in VR by operator C.

justifies ignoring the longitudinal aspects of the landmarking by modelling deviations between measurements as random errors.

For a specific specimen, annotation both in Stratovan Checkpoint and VR is based on the same 3D model, and digitization happened in the same local reference frame. Thus, the 48 landmark coordinate sets measured on the same specimen were directly comparable without first having to align them.

We further note that there is no ground truth landmark position in geometric morphometrics. For this reason, we focused on consistent landmark placement in our data analysis.

### Annotation time

We sorted the recorded annotation times by system and operator in the order the operators were annotating the skulls (Table S1), and computed trend lines using a linear model including quadratic terms. This allowed us to investigate differences in annotation time between systems and operators, and over time.

### Landmark-wise measurement error

We assessed landmark-wise annotation precision by computing the Euclidean distance between single landmark measurements and the corresponding landmark mean. In a first step, we visually compared the precision between systems and operators for each landmark. For that purpose, we computed the landmark means by averaging over replicas. In order to test landmark-wise whether medians across operators within one system were significantly different from each other, we used the `pairwisePercentileTest` function in the R package `rcompanion` (*Mangiafico, 2021*) to perform pairwise permutation tests based on 10,000 permutations. Additionally, we compared landmark-wise median overall precision between the two systems.

In a second step, to test whether the landmark-wise precision depends on the annotation system, we performed a three-way exploratory Analysis of Variance (ANOVA) with the factors *System*, *Operator* and *Specimen* separately for each landmark. Since we have a crossed data structure, we included all interaction terms. We note that our data are balanced, and that we only considered fixed effects models. Precision was computed from landmark means, which were obtained by averaging over replicas, systems and operators. Since the Euclidean distances between landmark measurements and means had right-skewed distributions, Euclidean distances were log-transformed to approximate a Gaussian distribution.

Since there is no ground truth involved in geometric morphometrics, we assessed accuracy by investigating how closely replicate landmarks were placed in VR compared to the traditional desktop system. For this purpose, we computed landmark means by averaging over replicas, followed by computing Euclidean distances between landmark means obtained from the two systems (for given operators and specimens). This allowed us to visually compare landmark-wise overall accuracy, and investigate differences in accuracy between operators for each landmark. For each landmark, we tested differences in operator median accuracy by performing pairwise permutation tests. We note, however,

that the statistical power of these tests is limited due to the small sample size of six annotated specimens per operator.

### Overall measurement error

We assessed overall measurement error similarly as previously described in *Messer et al. (2021)*. In a first step, we computed Procrustes distances between devices, between operators, and within operators (*i.e.* between landmark replica) to investigate overall measurement error. Here, we define Procrustes distance as the sum of distances between corresponding landmarks of two aligned shapes. This allowed us to investigate the extent of differences in the total shape of the same specimen in various ways: measurement by (a) the same operator using a different system (between-system error), (b) different operators using the same system (between-operator error), and (c) the same operator using the same system (within-operator error). Since all measurements from a specific specimen were in the exact same coordinate system, we did not have to align the landmark coordinates prior to the computation of Procrustes distances. Note that we computed Procrustes distances between all possible combinations, which introduces pseudoreplicates. For each error source, we tested differences in median Procrustes distances between operators, systems, or system-and-operator by performing pairwise permutation tests. We also compared median Procrustes distance between the error sources.

In a second step, we ran a Procrustes ANOVA (*Goodall, 1991*; *Klingenberg & McIntyre, 1998*; *Klingenberg, Barluenga & Meyer, 2002*; *Collyer, Sekora & Adams, 2015*) to assess the relative amount of measurement error resulting from the different error sources *System*, *Operator*, and *Specimen* simultaneously. With this approach, Procrustes distances among specimens are used to statistically assess the model, and the sum-of-squared Procrustes distances are used as a measure of the sum of squares (*Goodall, 1991*). As opposed to a classical ANOVA, which is based on explained covariance matrices, Procrustes ANOVA allowed us to estimate the relative contribution of each factor to total shape variation, which is given by the R-squared value. Prior to Procrustes ANOVA, the landmark configurations had to be aligned to a common frame of reference using a generalized Procrustes analysis (GPA) (*Gower, 1975*; *Ten Berge, 1977*; *Goodall, 1991*), in which the configurations were scaled to unit centroid size, followed by projection to tangent space. GPA eliminates all geometric information (size, position, and orientation) that is not related to shape. Since we have a crossed data structure, we used the following crossed model for the full Procrustes ANOVA: *Coordinates ~ Specimen × System × Operator*.

## RESULTS

### Annotation time

Figure 2 shows that all operators became faster at annotating a specimen over time, especially during the initial period of data collection. We observe that the two trend lines representing operator C's annotation times are exhibiting a minimum around annotation 20 (VR) and 25 (Stratovan Checkpoint). We point out that this is not an artefact due to
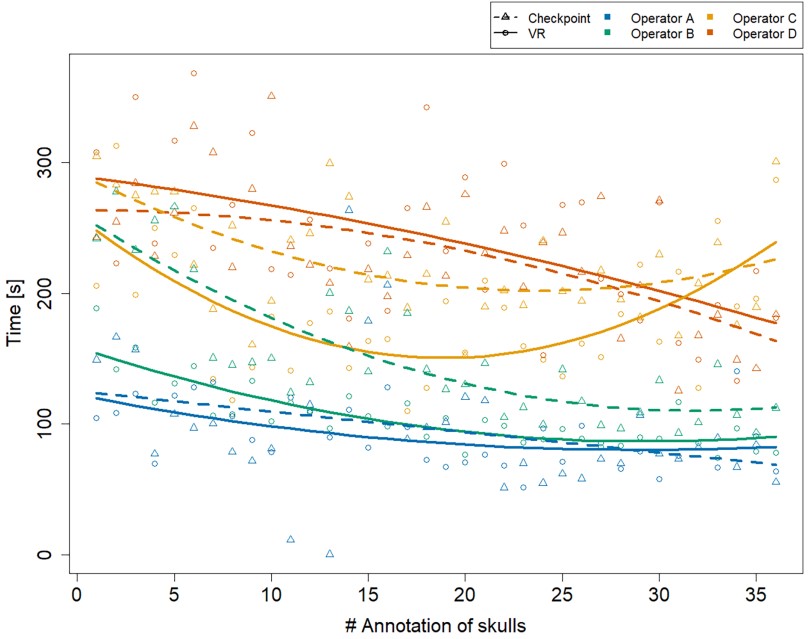

**Figure 2 Annotation time by system and operator over time.** The *x*-axis represents the skulls in the order the operators were annotating them. Trend lines suggesting a learning effect were estimated using a linear model including quadratic terms. Operator A's annotation time in Stratovan Checkpoint was not recorded properly for annotation 11 and 13.       

the quadratic trend, but that operator C's annotation times were actually increasing towards the end of data collection. This might be explained by the fact that C was the only operator collecting all data on the same day. The biologists A and D seemed to be equally fast in both systems over the whole data collection period. Operators B and C, that have a background in virtual reality, started out being much faster in VR, but were approaching VR annotation times in Stratovan Checkpoint over time.

## Landmark-wise measurement error

The boxplots of Euclidean distances between single landmark measurements and landmark means (Fig. 3) reveal that on an overall basis, a similar annotation precision was obtained for all six landmarks in VR compared to Stratovan Checkpoint. There were substantial differences between operators: Operator A, for example, was generally significantly less precise than the other operators. This might be explained by the fact that operator A, who is experienced in physical annotation, was not zooming in as much on the 3D models as the other operators during data collection. Moreover, operator B was significantly more precise in VR than other operators. Operators A and C appeared to be more precise in Stratovan Checkpoint compared to VR. Finally, operator D was much more consistent than the other operators, which is demonstrated by a similar obtained precision for all landmarks.

We obtained corresponding results in our ANOVA, which we ran separately for each landmark (Table 1): The factor *System* was only significant in case of landmark 11, but not for the five other landmarks. Computing the means of the Euclidean distance between

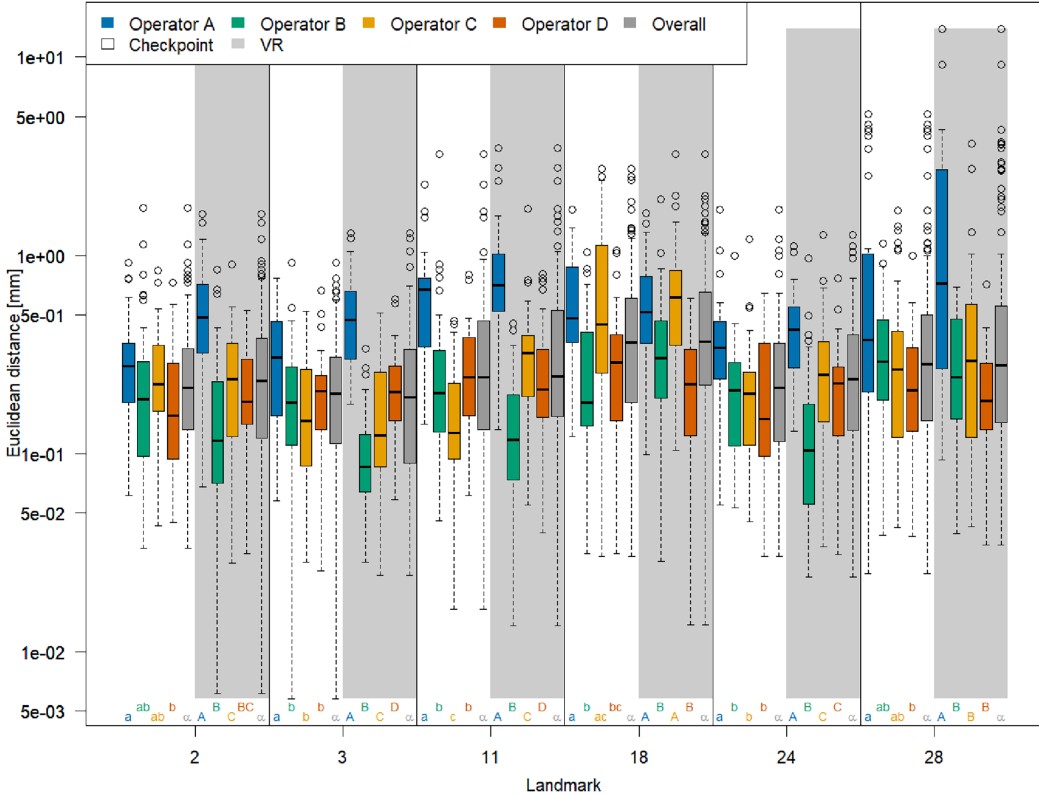

**Figure 3 Landmark-wise precision.** Precision is measured as the Euclidean distance between single landmark measurements and the operator landmark mean. For each landmark, we compared median overall precision between systems, and median precision between operators within each system for test of significance. For each of these 18 comparison rounds, groups sharing the same letter are not significantly different. The thick bars represent the median, boxes display the interquartile range, and the whiskers extend to 1.5 times the interquartile range. Circles represent outliers. Note that the vertical axis is logarithmic.

measurements and mean of landmark 11 separately for each system (VR: 0.57 mm; Stratovan Checkpoint: 0.69 mm) revealed that annotation of landmark 11 was more precise in VR compared to Stratovan Checkpoint. There was no strongly significant interaction between *System* and *Operator*, nor between *System* and *Specimen* for five landmarks (*System* and *Operator*: 2, 3, 11, 18, 24; *System* and *Specimen*: 2, 3, 11, 18, 28). As in Fig. 3, we detected major, significant differences between operators for all landmarks, which were expressed in large *F*-values. This was also true for interaction terms involving the factor *Operator*. Finally, we found that the variability in precision was larger between operators than between specimens, except for landmark 28. This exception can be explained by the fact that landmark 28 was subject to large outliers of type (3), measured by one operator (A) on one specimen, which were not excluded from the analysis. A similar effect is observed in Fig. 3. Examination of Q–Q-plots of the residuals showed that for most of the landmarks, the distribution of the residuals has heavier tails than the Gaussian distribution. However, since balanced ANOVAs are fairly robust to deviations from the Gaussian distribution, we decided not to investigate this further in this explorative study.

**Table 1 ANOVA, separately for each landmark.** Dependent variable is log-transformed Euclidean distance between single landmark measurements and landmark means. We applied the following crossed structure: System × Operator × Specimen. Residuals reflect landmark replica, and have 240 degrees of freedom.

| Variables | Df | F | Pr(>F) | F | Pr(>F) | F | Pr(>F) |
|---|---|---|---|---|---|---|---|
| | | LM 2 | | LM 3 | | LM 11 | |
| System | 1 | 2.32 | 0.129 | 0.54 | 0.461 | 9.53 | 0.002 |
| Operator | 3 | 21.15 | 0.000 | 76.02 | 0.000 | 57.56 | 0.000 |
| Specimen | 5 | 2.36 | 0.041 | 42.54 | 0.000 | 9.95 | 0.000 |
| System:Operator | 3 | 0.89 | 0.445 | 1.34 | 0.261 | 3.65 | 0.013 |
| System:Specimen | 5 | 2.16 | 0.059 | 0.43 | 0.825 | 3.01 | 0.012 |
| Operator:Specimen | 15 | 1.73 | 0.047 | 4.20 | 0.000 | 5.87 | 0.000 |
| System:Operator:Specimen | 15 | 0.82 | 0.657 | 2.45 | 0.002 | 3.01 | 0.000 |
| | | LM 18 | | LM 24 | | LM 28 | |
| System | 1 | 1.73 | 0.190 | 0.98 | 0.324 | 0.23 | 0.635 |
| Operator | 3 | 31.91 | 0.000 | 51.97 | 0.000 | 17.27 | 0.000 |
| Specimen | 5 | 21.60 | 0.000 | 12.32 | 0.000 | 183.35 | 0.000 |
| System:Operator | 3 | 0.42 | 0.741 | 2.01 | 0.113 | 7.69 | 0.000 |
| System:Specimen | 5 | 0.91 | 0.476 | 4.24 | 0.001 | 1.84 | 0.105 |
| Operator:Specimen | 15 | 2.64 | 0.001 | 6.18 | 0.000 | 8.01 | 0.000 |
| System:Operator:Specimen | 15 | 1.59 | 0.077 | 3.42 | 0.000 | 1.56 | 0.086 |

We note that we obtained corresponding *F*-values and significance levels when including outliers of type (2) in our ANOVA (Table S2). However, annotation of landmark 11 seemed to be more precise in Stratovan Checkpoint compared to VR (VR: 0.93 mm; Stratovan Checkpoint: 0.77 mm), which can be explained by the fact that the outlier of type (2) at landmark 11 was placed in VR.

With respect to accuracy (Fig. 4), we found that landmarks were generally placed at similar coordinates in both VR and Stratovan Checkpoint, for all landmarks. For our sample, the Euclidean distance between system means, averaged over specimens and operators, ranged from 0.165 mm (LM 3) to 0.465 mm (LM 28), which is of the same magnitude as the resolution of the 3D models. Similarly to precision, accuracy varied substantially between operators: In particular for landmarks 2, 11, and 28, operator A was less accurate than the other operators. However, this was only significant in case of landmark 2 when comparing the medians (based on the limited sample size of six specimens per operator). We note that the two specimens for which we had outliers of type (3), did not show the lowest accuracies for that particular landmark.

## Overall measurement error

The permutation significance test revealed that the median of the Procrustes distances within operators was not significantly different for both systems (Fig. 5), providing evidence that Stratovan Checkpoint and the VR annotation system exhibited similar precision for the group of operators participating in this study. The distribution of Procrustes distances between systems is comparable to that within operators, however, the

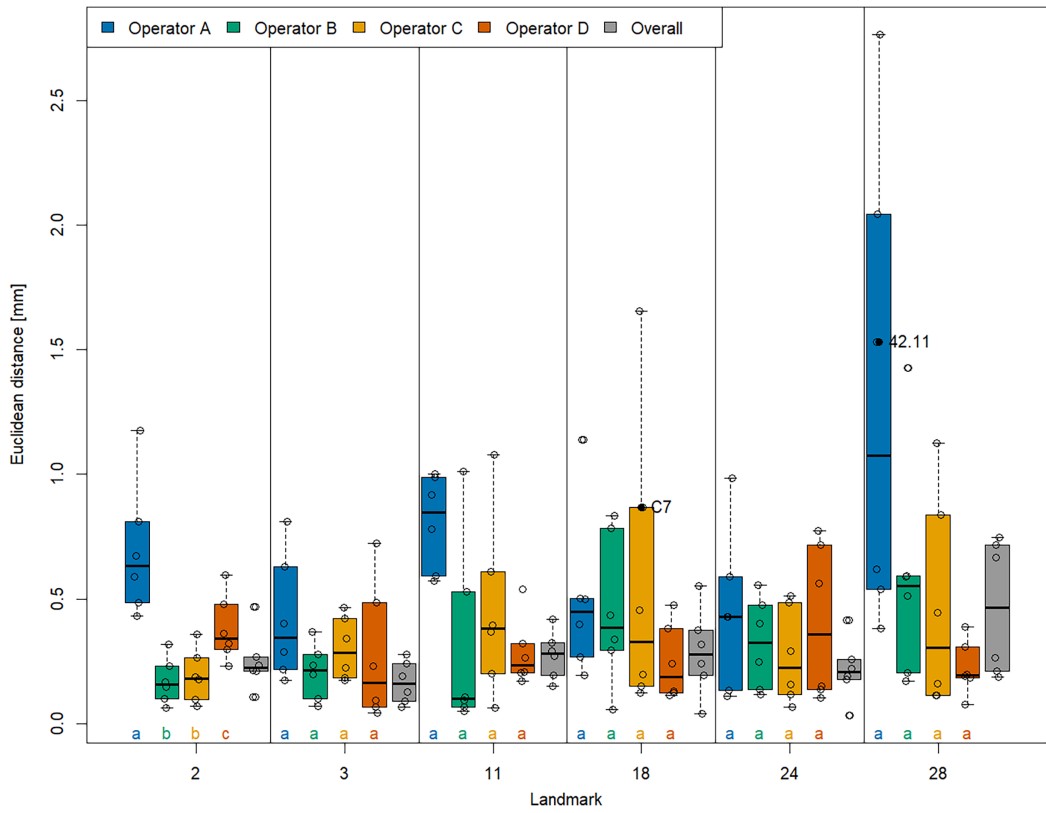

**Figure 4 Landmark-wise accuracy.** Accuracy is measured as the Euclidean distance between system means. For each landmark, median accuracies of operators sharing the same letter are not significantly different. Jittered data points correspond to the six specimens. The labelled two specimens correspond to the outliers of type (3). The thick bars represent the median, boxes display the interquartile range, and the whiskers extend to 1.5 times the interquartile range.

median of the Procrustes distances between systems is significantly larger than that within operators. In general, Procrustes distances between operators exhibited larger values than those between systems or within operators, with a significant difference in median, which validates the landmark-wise analysis. The median Procrustes distance between operators using the VR annotation system was significantly smaller than when using Stratovan Checkpoint. As in the landmark-wise analysis, we observed significant systematic differences between operators: For operators B and D, who were the only operators having experience in digitally placing landmarks, we found smaller measurement differences between systems than for operators A and C. A similar pattern was observed for within-operator error. Moreover, operator A was more precise in Stratovan Checkpoint, whereas operators B and D were more precise using the VR annotation system.

An analysis of the outliers revealed that they were almost exclusively measured on NHMD specimen 42.11 and FMNH specimen C7-98, where we observed outliers of type (3) (Fig. S3), and on NHMD specimen 664. The largest outliers are connected to measurements by operator A, and measurements in VR. Landmark 28 contributed

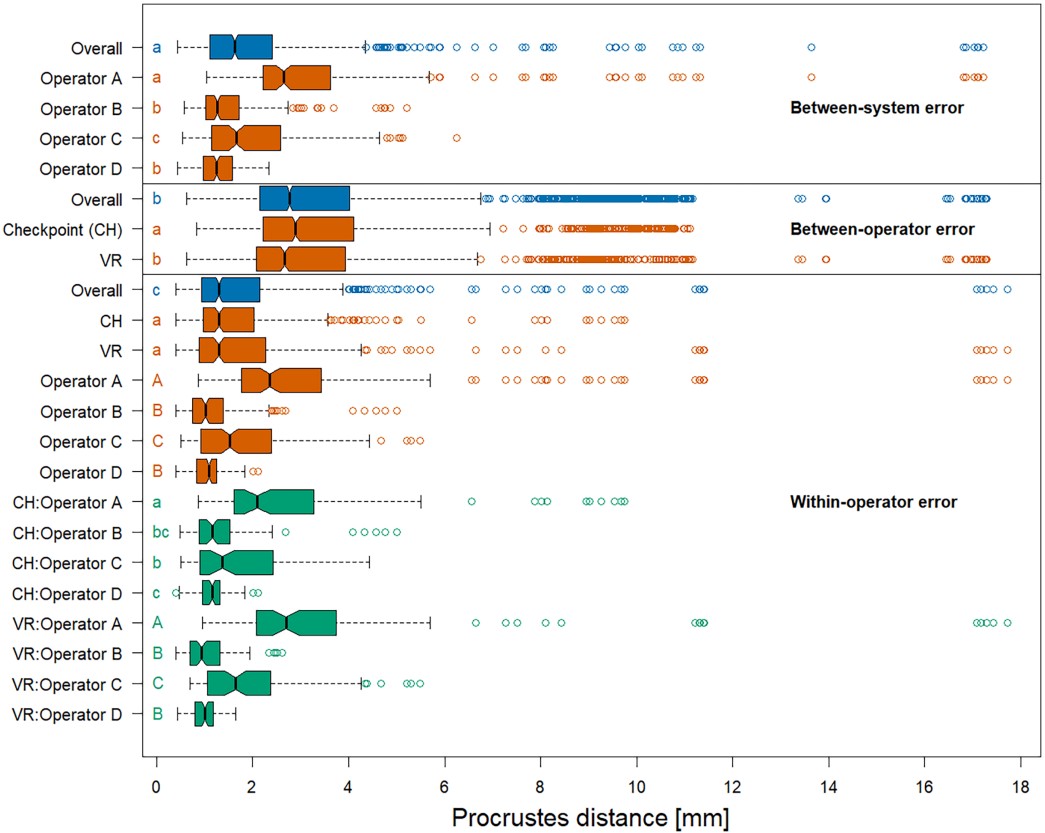

**Figure 5 Boxplots of Procrustes distances.** Computation of Procrustes distances between systems; between systems for a given operator; between operators; between operators for a given system; within operators; and within operators for a given system/operator/system-and-operator. Within an error source, we compared operator/system/system-and-operator medians with significance tests. Moreover, we compared median overall Procrustes distance between error sources. For each of these six comparison rounds, groups sharing the same letter(s) are not significantly different. The thick bars represent the median, boxes display the interquartile range, and the whiskers extend to 1.5 times the interquartile range. Outliers are represented by circles. The boxplot colours indicate whether a boxplot is based on all Procrustes distances for a given error source (blue), on a subset (red), or on a subset of a subset (green).

substantially to the large Procrustes distances. This is in line with the results on landmark-wise annotation precision (Fig. 3).

Running a Procrustes ANOVA on the whole dataset, again, validated the landmark-wise analysis (Table 2). The factor *System* did not seem to contribute much to total shape variation (0.04%), and a comparable result was obtained for the interaction terms involving the factor *System*. The main contributing factor of the two measurement error sources was *Operator*, which accounted for 1.6% of total shape variation. As in the landmark-wise ANOVAs, the interaction between *Operator* and *Specimen* is of importance and explained 1.7% of total shape variation, indicating that the operators were not experienced. As in Fig. 5, the results indicate that between-operator error was larger than between-system error. Most of the total shape variation (94.2%) was explained by

**Table 2 Procrustes ANOVA on shape.** We applied the following crossed structure: System × Operator × Specimen. Residuals reflect landmark replica. The R-squared values (Rsq) give estimates of the relative contribution of each factor to total shape variation.

| Variables | Df | MS | Rsq | F | Pr(>F) |
|---|---|---|---|---|---|
| System | 1 | 0.00035 | 0.0004 | 3.909 | 0.013 |
| Operator | 3 | 0.00527 | 0.0163 | 59.592 | 0.001 |
| Specimen | 5 | 0.18275 | 0.9421 | 2,065.941 | 0.001 |
| System:Operator | 3 | 0.00022 | 0.0007 | 2.526 | 0.003 |
| System:Specimen | 5 | 0.00012 | 0.0006 | 1.372 | 0.140 |
| Operator:Specimen | 15 | 0.00107 | 0.0166 | 12.114 | 0.001 |
| System:Operator:Specimen | 15 | 0.00010 | 0.0015 | 1.101 | 0.281 |
| Residuals | 240 | 0.00009 | 0.0219 | | |

biological variation among grey seal specimens. We note that our findings do not change when including outliers of type (2) in our Procrustes ANOVA (Table S3).

## DISCUSSION

We developed a VR-based annotation software to investigate whether VR is a viable alternative to mouse and keyboard for digital annotation of landmarks on 3D models. For this purpose, the VR annotation system was compared to the desktop program Stratovan Checkpoint as a tool for placing landmarks on 3D models of grey seal skulls. The two systems were compared in terms of overall and landmark-wise precision and accuracy, as well as annotation time. We used a carefully chosen setup, where four operators were placing six well-defined anatomical landmarks on six skulls in six trials, which allowed the investigation of multiple sources of measurement error (between systems, within and between operators, and between specimens).

On a desktop computer, an operator is forced to place landmarks through the point-of-view of their display. This is in contrast to VR, where an operator may annotate landmarks from angles different than their point-of-view, since the point-of-view is tracked using the head-mounted display and the controllers can be used to annotate landmarks from an arbitrary direction.

Another benefit of the VR system compared to Stratovan Checkpoint is that it allows the user to scale the specimen. Hence the application is agnostic to specimen size, which is especially helpful in annotating smaller specimens. In Stratovan Checkpoint, the specimen cannot be resized, but the camera can be placed closer. However, when placed too closely, the camera's near-plane will clip the specimen, thereby setting a limit on how closely one can view the specimen. In real-time rendering, two clipping planes are used to delimit the part of the scene that is drawn. The depth buffer has limited precision, and the greater the distance between these two planes, the more imprecise the depth buffer and the greater the risk of incorrectly depth sorted pixels. Unfortunately, the need to move the near plane away from the eye entails that if we move the camera very close to an object, it may be partially or entirely clipped by the near plane.

All in all, our analysis showed that annotation in VR is a promising alternative to desktop annotation. We found that landmark coordinates annotated in VR were close to landmark coordinates annotated in Stratovan Checkpoint. Taking mouse and keyboard annotation as the reference, this implies that landmark annotation in VR is accurate, which is in line with the findings of *Li et al. (2021)*. The accuracy achieved is of the same magnitude as the resolution of the 3D models. Furthermore, when investigating precision, both in landmark-wise ANOVAs, and a Procrustes ANOVA involving all landmarks at once, the factor *System* was not significant, in contrast to the factors *Operator* and *Specimen*. This demonstrated that the measured annotation precision in VR was comparable to mouse and keyboard annotation, whereas precision significantly differed between operators and specimens. These results are in line with previous studies on measurement error which found a larger between-operator compared to between-system or within-operator error (*e.g.*, *Shearer et al., 2017*; *Robinson & Terhune, 2017*; *Messer et al., 2021*). However, we obtained a much smaller between-system error when comparing annotation in VR with desktop annotation than *Messer et al. (2021)*, who compared physical and digital landmark placement on grey seal skulls.

Our results revealed that VR was significantly more precise than Stratovan Checkpoint for one landmark. A possible explanation is that the location of this landmark (no 11) on an apex along a margin (Fig. 1D) required an operator to observe the approximate location on a skull from various angles to decide on the landmark position. This might have been easier in VR because of the direct mapping between head and camera movement. The operators might also have benefited from the ability to look at landmark 11 independently of the angle used for landmark placement, allowing the operators to view the silhouette of the apex while pointing the annotation gun at the apex, perpendicular to the camera direction.

We found a weak indication that both precision in VR, and precision in general seemed to be positively linked to an operator's experience in placing landmarks on 3D models, and not necessarily to an operator's knowledge of VR or experience in placing landmarks on physical skulls. This result highlights the importance of annotation training on 3D models prior to the digital annotation process in order to obtain a higher precision. However, we have to keep in mind that the group of operators participating in this study is not a representative sample of professional annotators. Some of the operators in this study were not experienced in annotating landmarks. This was reflected in the interaction between *Operator* and *Specimen*, which was significant for all landmark-wise ANOVAs and in the Procrustes ANOVA.

We did not find any evidence that annotation in VR is faster compared to desktop annotation, which is in line with *Li et al. (2021)*. However, we did not include difficult to place landmarks, for which *Li et al. (2021)* found a significantly shorter annotation time in VR than on the desktop. Even though *Li et al. (2021)* conducted a similar experiment, there are three major differences to our setup: (1) In their study, the point the user was supposed to annotate was actually shown during data collection. This is different from the real-life annotation of landmarks we simulate, as the latter includes interpretation of the specimen's anatomy under the respective constraints and advantages of the two

interfaces. (2) They used an in-house 2D annotation tool, whereas our study involves an industry standard 2D annotation software. (3) In our study, the user manipulates the model with the non-dominant hand and annotates with the dominant hand. This is similar to how one adjusts the paper with the non-dominant hand and writes on it with the dominant hand. We compare this to Stratovan Checkpoint's unimanual interface where the mouse is used both for manipulation and annotation and the keyboard is used to change the mode of the mouse.

## CONCLUSIONS

To sum up, annotation in VR is a promising approach, and there is potential for further investigation. The current implementation of the VR annotation system is a basic prototype, as opposed to Stratovan Checkpoint, which is a commercial desktop software. Nevertheless, we did not find significant differences in precision between the VR annotation system and Stratovan Checkpoint. For one landmark, annotation in VR was even superior with respect to precision compared to mouse and keyboard annotation. Accuracy of the VR annotation system, which was measured relative to Stratovan Checkpoint, was of the same magnitude as the resolution of the 3D models. In addition, our study is based on a non-representative sample of operators, and did not involve any operator with a background both in biology and VR.

## FUTURE WORK

The VR controllers in our study are held in a power grip and do not provide haptic feedback although they can provide vibrotactile feedback (*i.e.* vibrate) when the user touches a surface. *Li et al. (2021)* employ the Geomagic Touch X which, as mentioned, is a precision grip kinesthetic device that does provide haptic feedback. Unfortunately, the haptic feedback comes at the expense of quite limited range since the pen is attached to an articulated arm. On the other hand, there are precision grip controllers which are not haptic devices and hence not attached to an arm. Thus, a future study comparing power grip, precision grip, and the combination of precision grip and haptic feedback would be feasible. Such a study might illuminate whether the precision grip or the haptic feedback is more important.

An interesting extension of the current VR system would be to add a kind of nudging to help the user make small adjustments to the placed landmarks. A further extension could be use of shape information through differential geometry to help guide annotation points toward local extrema.

Improvement of the VR system in terms of rendering performance would be beneficial as it would enable display of the 3D scan in all details (*Jensen et al., 2021*). This would potentially improve the user's precision when placing landmarks. Further improvements of the VR annotation system could be customizable control schemes, camera shortcuts to reduce annotation time, manipulation of the clipping plane to see hidden surfaces, rendering cross sections, and more UIs with information on the current annotation session.

In this study, we focused on clearly defined landmarks. It would be interesting to investigate (as in the work of *Li et al. (2021)*) whether VR might be superior to desktop annotation in the case of landmarks that are more difficult to place. Furthermore, most operators had no experience with one or both types of software. It would be very interesting with a longer term study to clarify the difference between the learning curves associated with the different annotation systems: how quickly does proficiency increase and when does it plateau? Such a study might also help illuminate whether habitually wearing an head mounted display is problematic. There is some fatigue associated with usage of a head mounted display, and this could become either exacerbated or ameliorated with daily use, something we could not address in this study.

## APPENDIX

**Table A1  List of original 3D models of grey seal (*Halichoerus grypus*) skulls used in this study and their source.** NHMD: Natural History Museum of Denmark; FMNH: Finnish Museum of Natural History. Skull length was approximated by the average Euclidean distance between landmarks 7 and 18 (Fig. 1) based on eight repeated measurements by *Messer et al. (2021)*.

| Institution | Specimen | Skull length (cm) | Source (MorphoSource identifiers) |
| --- | --- | --- | --- |
| NHMD | 42.11 | 19.3 | https://doi.org/10.17602/M2/M357658 |
| NHMD | 96 | 23.8 | https://doi.org/10.17602/M2/M364247 |
| NHMD | 223 | 21.3 | https://doi.org/10.17602/M2/M364279 |
| NHMD | 323 | 18.2 | https://doi.org/10.17602/M2/M364263 |
| NHMD | 664 | 21.5 | https://doi.org/10.17602/M2/M364287 |
| FMNH | C7–98 | 28.1 | https://doi.org/10.17602/M2/M364293 |

**Table A2  List of the six anatomical landmarks used in this study (L = left, R = right).** Four landmarks are of Type I, and two of Type II.

| Landmark description | Name | Type |
| --- | --- | --- |
| Caudal apex of nasal | 2 | I |
| Intersection of maxillofrontal sutureand nasal (L) | 3 | I |
| Anterior apex of jugal (R) | 11 | II |
| Dorsal apex of foramen magnum | 18 | II |
| Posterior point of last molar (L) | 24 | II |
| Ventral apex of orbital socket (L) | 28 | II |

## ACKNOWLEDGEMENTS

The authors would like to thank Daniel Klingberg Johansson from the Natural History Museum of Denmark, and Mia Valtonen from the Institute of Biotechnology at University of Helsinki for giving us access to the grey seal collections. We would further like to thank Florian Gawrilowicz for his assistance with the 3D model creation part, and for providing us with his code for non-rigid point cloud registration.

### Funding

The authors received no funding for this work.

### Competing Interests

The authors declare that they have no competing interests.

### Author Contributions

- Dolores Messer conceived and designed the experiments, performed the experiments, analyzed the data, prepared figures and/or tables, authored or reviewed drafts of the paper, and approved the final draft.
- Michael Atchapero conceived and designed the experiments, performed the experiments, authored or reviewed drafts of the paper, designed and implemented the VR annotation system, and approved the final draft.
- Mark B. Jensen conceived and designed the experiments, performed the experiments, authored or reviewed drafts of the paper, designed the VR annotation system, and approved the final draft.
- Michelle S. Svendsen performed the experiments, authored or reviewed drafts of the paper, and approved the final draft.
- Anders Galatius conceived and designed the experiments, performed the experiments, authored or reviewed drafts of the paper, and approved the final draft.
- Morten T. Olsen conceived and designed the experiments, authored or reviewed drafts of the paper, and approved the final draft.
- Jeppe R. Frisvad conceived and designed the experiments, authored or reviewed drafts of the paper, designed the VR annotation system, and approved the final draft.
- Vedrana A. Dahl conceived and designed the experiments, authored or reviewed drafts of the paper, and approved the final draft.
- Knut Conradsen conceived and designed the experiments, authored or reviewed drafts of the paper, and approved the final draft.
- Anders B. Dahl conceived and designed the experiments, authored or reviewed drafts of the paper, and approved the final draft.
- Andreas Bærentzen conceived and designed the experiments, authored or reviewed drafts of the paper, designed the VR annotation system, and approved the final draft.

### Data Availability

The landmark data is available at DTU Data: Messer, Dolores; Atchapero, Michael; Jensen, Mark Bo; Strecker Svendsen, Michelle; Galatius, Anders; Tange Olsen, Morten; et al. (2021): 3D Landmark data on grey seal skull measured on 3D surface models using the Stratovan Checkpoint software, and a VR annotation system. Technical University of Denmark. Dataset. https://doi.org/10.11583/DTU.14977353.v1.

The 3D models (not downsampled) are available at MorphoSource (Table A1).

## Supplemental Information

Supplemental information for this article can be found online at http://dx.doi.org/10.7717/peerj.12869#supplemental-information.

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
