# Peer review of "Using virtual reality for anatomical landmark annotation in geometric morphometrics"

_PeerJ, doi:10.7717/peerj.12869_

## Round 0.1 · original submission · Major Revisions

The reviewers pointed out some key shortcomings of the paper and should be addressed before further consideration. Please provide a response letter and the revised version with tracked changes. Thanks.

Reviewer 1 ·

Basic reporting

1. The authors should make the result section more concise and discuss the results only in discussion section. In current form, the result section is too long and hard to follow.
2. The figure 2 is hard to understand, there is too many points and the colours are not well perceived. I suggest simplifying the figure. Do the regression lines suggest learning effect?
3. Statistical significant differences in figure 3, 4, 5 should be indicated in the figures.

Experimental design

1. The number of included operators is low. This should be emphasized in discussion section.
2. When annotating with traditional system, did authors use the simultaneous view three perpendicular cross sections?

Validity of the findings

1. The authors included only four operators which significantly limits the generalizability of their findings.

Additional comments

1. The authors included a video of virtual reality system in the supplement. Can a short video/figures of the traditional system using 2D display and mouse and keyboard also be provided? This would allow readers that are not familiar with such system to better understand it.
2. Could the accuracy of annotation using virtual reality be improved if using haptic device that enables precise movements and has six degrees of freedom, instead of using classical VR controller with power grip.
3. In your virtual reality, is it possible to also “look” through the model and look at its cross section? Could such view improve the landmark recognition and annotation accuracy?

Reviewer 2 ·

Basic reporting

The authors present a study comparing two systems for annotating landmarks on 3D models, in this case seal skulls. One is a virtual reality system that uses a stereoscopic headset, the other is a conventional system with a monitor and mouse. The topic is of some interest and the manuscript is well written with good illustrations and adequate references. However, the title of the manuscript is quite general. It should describe the content a little more precisely. In Fig. 3, the positions of the magnified details in the images of the skulls should be marked.

Experimental design

It remains unclear why the resolution of the 3D model was also reduced for the measurement with the Stratovan Checkpoint software. This may not take into account an important advantage of this method.

Validity of the findings

To put the accuracy achieved into perspective, the resolution of the 3D models studied should also be provided in physical units. For this purpose, e.g. the mean distance of the vertices of the model could be given.
In my own experience, wearing a VR headset for extended periods of time can be quite exhausting. The authors should address the issue in the discussion.

Additional comments

The authors write "Procrustes distance is defined as the sum of distances between corresponding landmarks of two aligned shapes". Various definitions of the Procrustean distance exist. The authors may want to check if this describes the version used in the "geomorph" package.

Reviewer 3 ·

Basic reporting

The paper has a clear aim and objectives but suffered from no ground truth to support their analysis on accuracy. Also there are several issues need to be clarified.

Experimental design

There are several issues in the paper which are needed to be clarified:
1. There is no ground truth in the study that no evidence to support that the VR is more accurate.
2. Type 2 and 3 outliers should be replaced and at least the authors needs to show the results without replacing outliers.
3. 6 landmarks of 6 skulls and marked 6 times of each landmarks on 2 systems, it would be 432 rather than 288 landmark configurations; and its is not clear that how the authors handle the repeated measurements.
4. It is not clear how the mean landmarks were calculated.
5. Why GPA instead of Partical Procrustes Analysis (PPA) was used to assess the measurement errors? I think that the Procustest distance was scaled down by the centroid size.

Validity of the findings

VR for landmarking has been promoted for three decades but its benefits were limited in clinical practices. The study did not have ground truth to support the analysis that it is not reliable to state that VR is more accurate (maybe it is true).

---

## Round 0.2 · Minor Revisions

A minor revision is needed before further processing. I look forward to receiving your revision.

Reviewer 1 ·

Basic reporting

The authors satisfactorily adressed all my objections.

Experimental design

The authors satisfactorily adressed all my objections.

Validity of the findings

The authors satisfactorily adressed all my objections.

Reviewer 2 ·

Basic reporting

In my opinion, the authors have adequately addressed the issues mentioned.

Experimental design

No commment

Validity of the findings

No comment

Reviewer 3 ·

Basic reporting

no comment

Experimental design

Partial Procustes Analysis should be used instead of General Procurstes Analsysis since there should be no scale but only rotations and translations in the shape aligning or calculating the mean shape. Scaling in General Procustes Alignment would change the size of the shapes to be aligned.

Validity of the findings

The authors did not address the issue of the repeated measurements. The repeated data sampled from the same sample 6 times were correlated.

---

## Round 0.3 · accepted · Accept

All comments have been addressed. I recommend it for publication.